# Modular ketal-linked prodrugs and biomaterials enabled by organocatalytic transisopropenylation of alcohols

Na Yu[1,2,5], Yang Xu[1,5], Tao Liu[1], Haiping Zhong[1], Zunkai Xu[1], Tianjiao Ji[3], Hui Zou[1], Jingqing Mu[1], Ziqi Chen[1], Xing-Jie Liang 🔵 [2,4], Linqi Shi[1], Daniel S. Kohane 🔵 [3✉] & Shutao Guo 🔵 [1✉]

Isopropenyl ethers are critical intermediates for accessing medicinally valuable ketal-based prodrugs and biomaterials, but traditional approaches for the synthesis of isopropenyl ethers suffer from poor functional group compatibility and harsh reaction conditions. Here, we develop an organocatalytic transisopropenylation approach to solve these challenges, enabling the synthesis of isopropenyl ethers from various hydroxyl-group-containing small-molecule drugs, polymers, and functional building blocks. The method provides a straight-forward and versatile synthesis of isopropenyl ethers, features excellent tolerance of diverse functional groups, applies to a wide range of substrates, and allows scalable synthesis. The development of this organocatalytic transisopropenylation approach enables access to modular preparation of various acid-sensitive ketal-linked prodrugs and functionalized ketalated biomaterials. We expect our syntheses and transformations of isopropenyl ethers will find utility in several diverse fields, including medicinal chemistry, drug delivery, and biomaterials.

[1] Key Laboratory of Functional Polymer Materials of Ministry of Education, State Key Laboratory of Medicinal Chemical Biology and Institute of Polymer Chemistry, College of Chemistry, Nankai University, Tianjin 300071, P.R. China. [2] Translational Medicine Center, Key Laboratory of Molecular Target & Clinical Pharmacology, School of Pharmaceutical Sciences & The Second Affiliated Hospital, Guangzhou Medical University, Guangzhou 510260, P. R. China. [3] Laboratory for Biomaterials and Drug Delivery, Division of Critical Care Medicine, Children's Hospital Boston, Harvard Medical School, 300 Longwood Avenue, Boston, MA 02115, USA. [4] CAS Key Laboratory for Biological Effects of Nanomaterials and Nanosafety, National Center for Nanoscience and Technology, Beijing 100190, P.R. China. [5] These authors contributed equally: Na Yu, Yang Xu. ✉email: daniel.kohane@childrens.harvard.edu; stguo@nankai.edu.cn

Many environments within the human body are acidic, such as the stomach, tumors, inflamed tissues, and acidic organelles (such as endolysosomes)[1]. This acidity has been used to provide pH-dependent drug release, to improve efficacy and the therapeutic index[2–4]. Recently, isopropenyl ethers (IPPEs) have emerged as critical intermediates to produce medicinally valuable asymmetric acyclic ketals that are highly acid-sensitive and derived from the metabolite acetone, and therefore are useful for the development of innovative stimulus-responsive drug delivery systems, including prodrugs for a large subset of hydroxyl-group-containing drugs[5–13]. Besides, IPPEs can be involved in many other transformations, including Diels-Alder reaction and construction of $C(sp^3)$-rich $\alpha$-tertiary dialkyl ethers[14–21]. Given that many small-molecule drugs (e.g., taxanes, sterols, prostaglandins, and nucleosides) contain a hydroxyl group, transisopropenylation of alcohols to create IPPEs can be expected to be useful in late-stage derivatization of these drugs. For instance, drug-derived IPPEs could be used to synthesize acyclic-ketal-linked prodrugs modularly for comprehensive optimization of prodrug performance. In the context, we hypothesized that a facile method for versatile synthesis of IPPEs with a broad substrate scope—including hydroxyl-group-containing drugs—would be highly desirable, with the goal of facilitating the development and application of medicinally valuable ketals.

Several methods are reported to synthesize IPPEs from alcohols or their derivatives: for example, acid-catalyzed pyrolysis of alkoxypropanes[22–24]; treatment of 2-methoxypropyl (MOP) ketal of sterically hindered alcohols with an excess of trimethylsilyl triflate (TMSOTf) and base[25]; olefination of acetates using Tebbe reagents[26,27] and reaction of alcoholic potash with methylacetylene[28]. However, these reactions are usually carried out under harsh reaction conditions (e.g., high temperatures, use of strong bases), and some methods suffer from requiring expensive and toxic organometallic reagents and poor functional group compatibility (Fig. 1a). Numerous attempts, including transition-metal-catalyzed addition of alcohols to alkynes and cross-coupling reaction[29–37], have been made to complement approaches mentioned above. Nonetheless, no methods—which are compatible with complex drugs (e.g., paclitaxel, PTX) and various functional groups (such as acrylate, ester, and maleimide) that are either present in some biodegradable polymers or useful for the development of ketal-based drug delivery systems—have appeared so far.

Here, we report a straightforward transisopropenylation approach to the synthesis of IPPEs from alcohols and 2-methoxypropene (2-MPE), using a Brønsted acid–base pair as the organocatalyst (Fig. 1b). The method features mild reaction conditions, readily available and cheap starting materials and catalysts, easy workup, and a wide range of substrates; it is also compatible with complex small-molecule drugs, some polymers, and various functional groups and allows scalable synthesis. The transisopropenylation approach can facilitate the facile synthesis of various asymmetric ketal-linked prodrugs and functionalized ketalated biomaterials. An IPPE derivative of PTX is applied as the model IPPE to demonstrate the utility of drug-derived IPPEs for the modular preparation of acetone-based acyclic-ketal-linked prodrugs, including heterodimeric prodrugs containing both PTX and floxuridine (FUDR), two anticancer agents with different tumor-inhibiting mechanisms.

## Results and discussion
**Design rationale.** Given that there are many small-molecule hydroxyl-group-containing drugs, a method for synthesizing IPPEs from alcohols using a mild catalytic transisopropenylation strategy might be useful. However, such catalytic

transisopropenylation approaches have been scarcely reported[38]. Although there are several transition-metal-catalyzed methods for transvinylation of alcohols[38–43], most of them do not apply to transisopropenylation of alcohols[25,42,44]. To the best of our knowledge, the only catalytic transisopropenylation of alcohols was achieved using isopropenyl acetate and $[Ir(cod)Cl]_2/Na_2CO_3$ catalytic system[38]. Still, the method's substrate scope is limited by harsh reaction conditions (high temperature).

We noted that resonance-stabilized oxocarbenium ion equilibrates with ketal under acidic conditions[45], and that the rearrangement of oxocarbenium ion can proceed to afford IPPEs in the presence of weak bases[25]. Thus, we envisioned that a Brønsted acid–base pair could be used as an organocatalyst to not only catalyze the reaction between alcohol and the IPPE 2-MPE to form MOP ketal, but also catalyze the oxocarbenium ion in the equilibrium to undergo rearrangement to provide new IPPE. Therefore, a simple, mild, and catalytic transisopropenylation approach to syntheses of IPPEs could be developed (Fig. 1b).

**Reaction development.** We started by carrying out the reaction of alcohols and 2-MPE (**2a**) in THF using 2-octanol (**1a**) as the substrate (Fig. 1c and Supplementary Tables 1–6). According to the assumption, Brønsted acid can catalyze alcohol and **2a** to give MOP ketal, so we evaluated Brønsted acids with different acidity firstly. An excess of cheap **2a** (16 equiv.) was used to drive the equilibrium toward the oxocarbenium ion production. Surprisingly, we found that the reaction using p-toluenesulfonic acid (pTSA) produced **3a**. However, it also produced a significant amount of side products that were dimer and oligomers of **2a** (Supplementary Figs. 1–2), caused by acid-initiated cationic polymerization of **2a**. Because base plays a vital role in promoting the oxocarbenium ion in the equilibrium to undergo rearrangement to afford IPPE, we selected pTSA as the Brønsted acid for acid–base pair and prepared several p-toluenesulfonate catalysts to evaluate bases. As a result, p-toluenesulfonate catalysts notably inhibited the side reactions, and the yield of **3a** was markedly affected by the bases' strength. Overall, 2,6-lutidinium p-toluenesulfonate (LPTS) gave the most satisfactory results.

**Synthetic scope.** With the optimized reaction condition, reactions of **2a** with various drug substrates were then examined (Fig. 2; see details in Supplementary Information). We started by carrying out the reaction using PTX (a diol with potent anticancer activity). We found that the reaction produced a mixture of PTX-diMOP and PTX-2′-MOP-7-IPPE and that the 7-position but not the 2′-position of PTX was selectively transisopropenylated (Supplementary Fig. 3). As PTX could not be selectively converted to the diIPPE product, instead a mixture of MOP and IPPE, 2′-hydroxyl of PTX was first protected by TBS (tert-butyldimethylsilyl), and then 7-hydroxyl of PTX was selectively transisopropenylated. As a result, we successfully synthesized PTX-2′-TBS-7-IPPE (**3b**) from PTX-2′-TBS with a yield of 85%. To the best of our knowledge, this is the first report of drug-derived IPPEs. However, no PTX-derived IPPEs were yielded when we tried traditional approaches (such as $[Ir(cod)Cl]_2/Na_2CO_3$ catalysis method and TMSOTf-promoted dealkoxylation) (Supplementary Figs. 13–16). Note that we ruled out Tebbe methylenation because Tebbe reagent is nonselective and would have decomposed the PTX.

We further demonstrated the versatility of this method by synthesizing IPPEs from more drugs: the diols fulvestrant (FUL, a sterol), FUDR (a nucleoside analogue), and tafluprost (TAF, a prostaglandin), as well as the monoalcohols zidovudine (ZDV, an anti-HIV agent), lovastatin (LOV, a statin), idebenone (IDBN, an anti-Alzheimer agent), podophyllotoxin (PPT, an anticancer

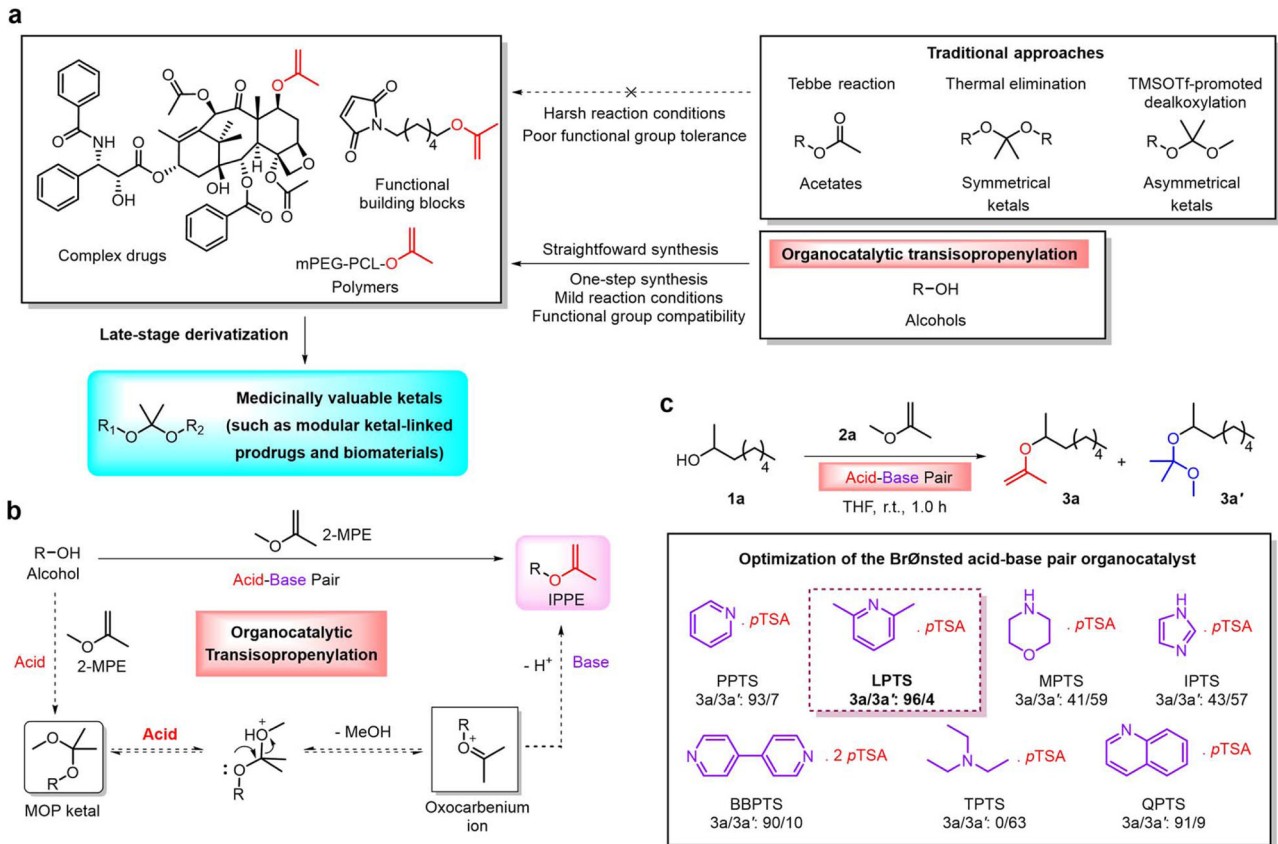

**Fig. 1 Design of an organocatalytic transisopropenylation approach for the synthesis of IPPEs from alcohols. a** Organocatalytic transisopropenylation enables the straightforward synthesis of IPPEs from hydroxyl-group-containing complex drugs (e.g., PTX), functional building blocks, and polymers under mild reaction conditions, complementing traditional approaches that are limited by the poor functional group compatibility and harsh reaction conditions. The resulting IPPEs will facilitate the synthesis of medicinally valuable ketals, such as modular ketal-linked prodrugs and biomaterials. **b** Organocatalytic transisopropenylation of alcohols, using 2-MPE with catalysis by Brønsted acid–base pair. Alcohol is transisopropenylated via the MOP ketal and transient oxocarbenium ion. **c** Optimization of the Brønsted acid–base pair organocatalyst. General conditions: **1a** (1.0 mmol, 1.0 equiv., 0.5 M), **2a** (16.0 equiv.), 0.5 mol % catalyst, THF, 25 °C, 1.0 h. The conversion rate of **1a** and molar ratio of **3a** to **3a'** were determined by NMR.

agent), testosterone (TEST, an anticancer agent), and abiraterone (ABI, an anticancer agent). The diols were first treated with **2a**, and then the products were analyzed by means of NMR spectroscopy to ascertain the sites of IPPE and MOP ketal formation. As a result, for the anticancer agent FUL, the IPPE formed at the secondary hydroxyl group (17-OH), whereas a MOP ketal formed at the phenol group (3-OH) (Supplementary Fig. 4). Similar to PTX, FUL could be not be selectively converted to the diIPPE product. Therefore, TBS protection of FUL at 3-position was first done, and then FUL-3-TBS-17-IPPE (**3c**) was synthesized from FUL-3-TBS. For FUDR and TAF, both hydroxyl groups could be converted to IPPEs (Supplementary Figs. 5–6). For drugs with two IPPE conversion sites, they could be either mono TBS-protected or used as is for the preparation of drug-derived IPPEs. For example, an IPPE derived from the anticancer agent FUDR, designated FUDR-5′-TBS-3′-IPPE (**3d**), was synthesized from FUDR-5′-TBS. In contrast, either no reaction was observed or decomposition occurred when we attempted to use [Ir(cod)Cl]$_2$/Na$_2$CO$_3$ catalysis method and TMSOTf-promoted dealkoxylation to prepare FUDR-derived IPPEs. Besides, the antiglaucoma drug TAF was directly converted to a diIPPE, designated TAF-3,5-diIPPE (**3e**). For the monoalcohols, ZDV, LOV, ABI, PPT, TEST, and IDBN were transformed to ZDV-IPPE (**3f**), LOV-IPPE (**3g**), ABI-IPPE (**3h**), PPT-IPPE (**3i**), TEST-IPPE (**3j**), and IDBN-IPPE (**3k**), respectively. We also explored the method's applicability to additional drugs with

tertiary alcohols and/or aromatic alcohols (such as 7-ethyl-10-hydroxycamptothecin, 7-hydroxycoumarin, and estradiol). However, we found that the hindered tertiary alcohol did not react at all due to steric hindrance (Supplementary Figs. 7–8) and that phenols gave only the corresponding MOP ketals (Supplementary Figs. 7–11). Nevertheless, the phenol of vitamin E with electron-donating groups afforded vitamin E-IPPE (**3l**). These results indicate that our method for the synthesis of drug-derived IPPEs from hydroxyl-group-containing drugs has a broad substrate scope, and it has the potential to be useful for the synthesis of IPPEs derived from analogues of these drugs.

Having studied the drug scope exclusively, we probed this strategy for alcohols widely used in pharmaceutical science and biomaterials. To this end, two kinds of hydroxyl-bearing polymers (**1m**, mPEG-OH, $M_n$ = 132, 550, 2000 g/mol; **1n**, mPEG-PCL-OH, $M_n$ = 2000–1800 g/mol), cholesterol **1o**, oleyl alcohol **1p**, and menthol **1q** were subjected to this protocol. To our delight, polymers were successfully converted to corresponding IPPEs (**3m** and **3n**), enabling the potential synthesis of acid-sensitive macromolecular ketal-linked conjugates. Moreover, we probed this strategy for alcohols that are widely used as functional building blocks. Various functional groups, including ester, halogen, acrylate, alkenyl, alkynyl, amide, benzyl, epoxy, nitryl, cyano, and maleimide (Fig. 2 and Supplementary Table 7), were well tolerated. These functional IPPEs may have broad applications in post-modification of ketal-based materials[5,46–52]. Besides,

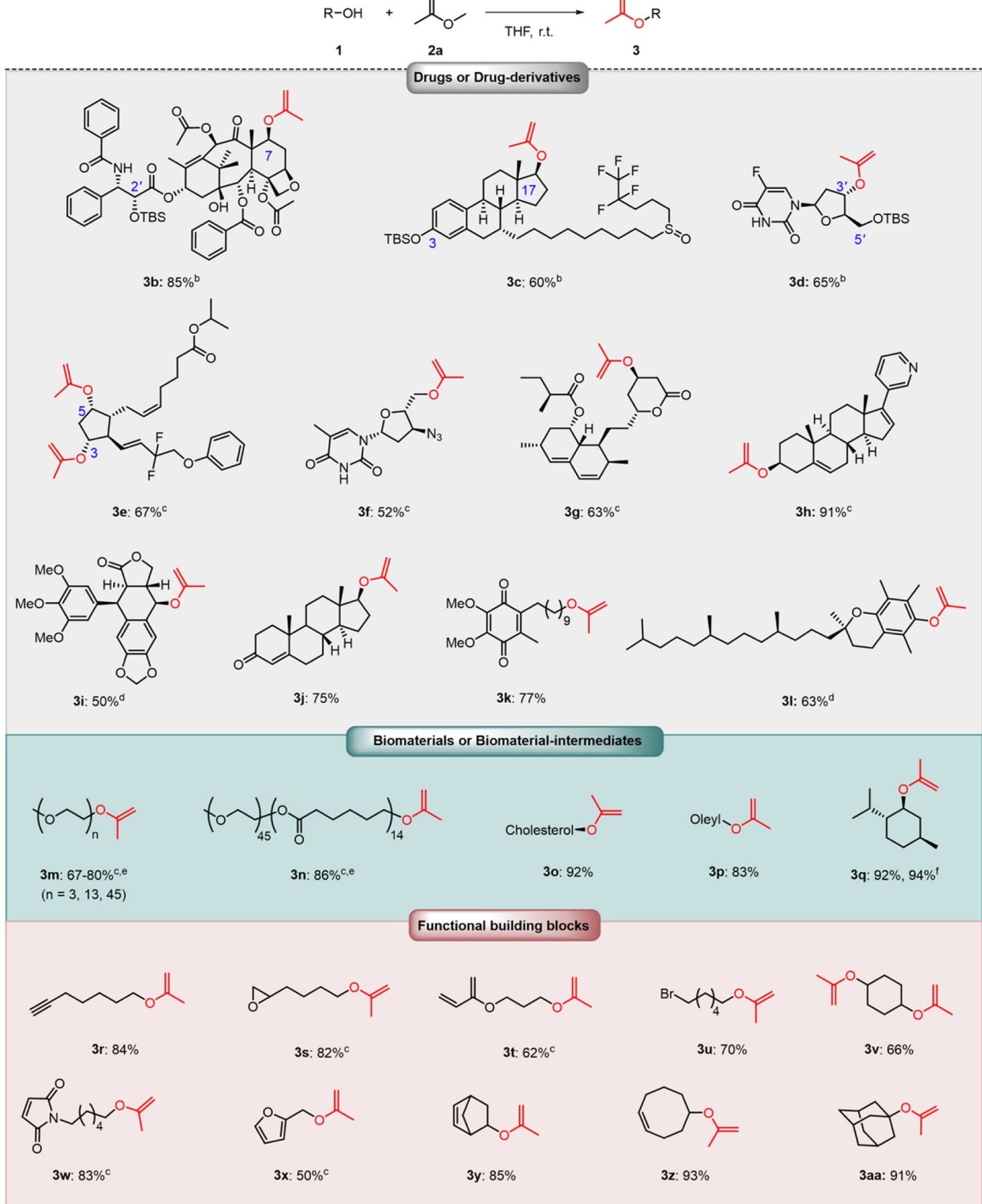

**Fig. 2 Synthetic scope of alcohols[a].** [a]Reaction conditions: **1** (1.0 equiv.), **2a** (16.0 equiv.), 0.5 mol % LPTS, THF, 25 °C, 12 h. [b]20.0 mol % LPTS, 48 h. [c]2.0 mol % LPTS, 24 h. [d]5.0 mol % LPTS, 48 h. Yields of isolated products are given. [e]Purity of **3** is provided, as determined by [1]H NMR. [f]Large scale synthesis: 50 mmol **1**, isolated yields by distillation.

**Fig. 3 Synthetic scope of enol ethers[a].** [a]Reaction conditions: **1j** (1.0 equiv.), **2** or **4** (16.0 equiv.), 20.0 mol % LPTS, THF, 25 °C, 24 h. Yields of isolated products are given.

the method was amenable to other primary, secondary, and tertiary alcohols (Supplementary Table 7), with good to high yields. The scalability of our approach was verified using **1a** and **1q** on 50 mmol scale, and large quantities of **3a** and **3q** (yield > 85%) were obtained by distillation.

Encouraged by the above results, we further probed whether other enol ethers were subjected to this method (Fig. 3), and we found various 2-methoxy-1-alkenes reacted with drug **1j** to produce corresponding enol ethers (**5a–f**), which would allow the synthesis of ketals with adjustable hydrolysis kinetics. Likewise, 2-methoxy-1-alkenes also reacted with other alcohols (such as primary alcohols **1k** and **1w**, and secondary alcohol dehydroepiandrosterone) to afford corresponding enol ethers in good yields (Supplementary Table 8).

The monitoring reaction process of **2a** with alcohol revealed that alcohol was firstly converted to MOP ketal, and then MOP ketal was transformed to new IPPE (Supplementary Fig. 17a). Besides, isolated MOP ketal was reacted with **2a** or 2-ethoxypropene under similar conditions to yield the same IPPE (Supplementary Fig. 17b), further confirming the proposed mechanism of the method (Supplementary Fig. 18). Other enol ethers should also be formed according to similar mechanisms.

**Synthesis of modular acetone-based ketal-linked prodrugs.** The use of prodrugs—in which a drug is covalently but reversibly linked to another moiety and can be released in its active form—

has emerged as an essential strategy for modifying drug performance[3,53,54]. Because many diseased tissues are characterized by low pH, acid sensitivity would be a desirable characteristic of such prodrugs[55]. Therefore, acyclic ketals, which are more acid-labile than analogous acetals and cyclic ketals, have great potential for the development of acid-activatable, traceless prodrugs of hydroxyl-group-containing drugs[11,12,56,57]. However, the synthesis of acyclic-ketal-linked prodrugs is markedly hindered by a lack of effective synthetic methods, and only a few acyclic-ketal-linked prodrugs have been reported[5,7–9,49]. Given that IPPEs could readily react with hydroxyl-group-containing drugs to afford acyclic-ketal-linked prodrugs and having established the organocatalytic transisopropenylation method for synthesis of IPPEs, we further explored the utility of this method for accessing divergent asymmetric ketal-linked prodrugs (Fig. 4 and see details in Supplementary Information). As proof-of-principle, hydroxyl-group-containing dexamethasone (DEX), a hydrophobic drug, was reacted with a PEG-derived IPPE **3m** ($M_n = 2000$ g/mol) under catalysis by dichloroacetic acid (DCA) to obtain a water-soluble $O,O$-ketal-linked PEGylated prodrug, i.e., PEG-K-DEX (Fig. 4a). In addition to $O,O$-ketal-linked prodrug, an $O,S$-ketal-linked prodrug and an $O,ON$-ketal-linked prodrug, which both are rarely reported and may have different stimuli-responsiveness than $O,O$-ketals[58–60], were also enabled by using a lauryl alcohol-derived IPPE under similar reaction conditions. Specifically, an $O,S$-ketal prodrug of thiol-group-containing captopril methyl ester (CAPME) and an $O,ON$-ketal prodrug of hydroxylamine-

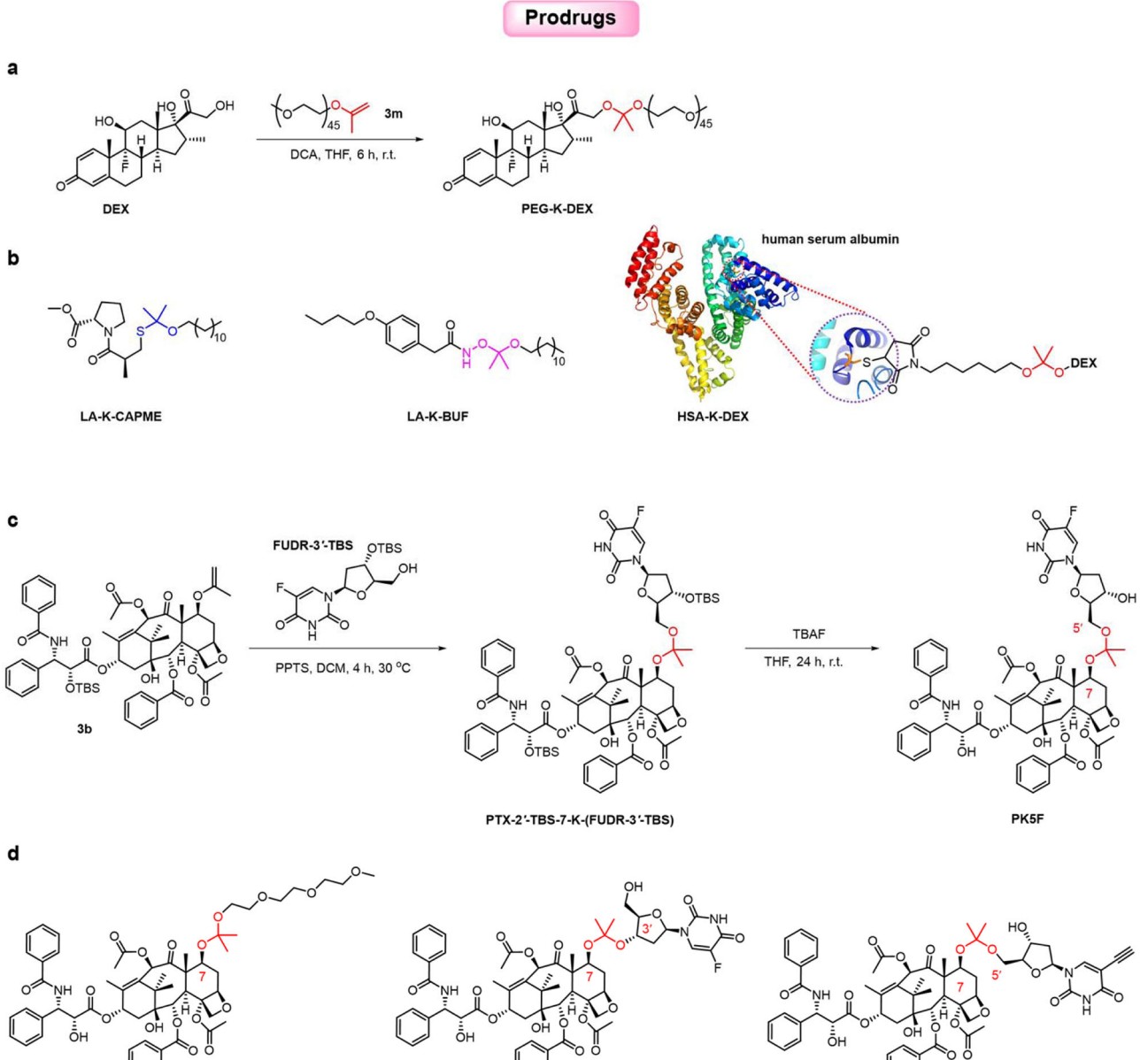

**Fig. 4 Ketal-linked prodrugs. a** synthesis of PEG-K-DEX; **b** structures of LA-K-CAPME, LA-K-BUF, and HSA-K-DEX; **c** synthesis of PK5F; **d** structures of PTX-7-K-EG₃, PK3F, and PK5E.

group-containing bufexamac (BUF) (Fig. 4b) were synthesized. Moreover, IPPE **3w** was similarly reacted with hydroxyl-group-containing DEX to obtain an intermediate, which was subsequently conjugated to human albumin serum (HSA) through maleimide-thiol reaction to yield a water-soluble *O,O*-ketal-linked protein prodrug conjugate, i.e., HSA-K-DEX (Fig. 4b and Supplementary Fig. 19).

To demonstrate the utility of drug-derived IPPEs for the modular preparation of ketal-linked prodrugs, we further synthesized four PTX prodrugs, designated PK5F, PTX-7-K-EG₃, PK3F, and PK5E (Fig. 4c, d). As shown in Fig. 4c, PK5F was synthesized using PPTS-catalyzed reaction of PTX-2′-TBS-7-IPPE **3b** with FUDR-3′-TBS and subsequent removal of the TBS group with tetra-n-butylammonium fluoride (TBAF). Similarly, PTX-7-K-EG₃, PK3F, and PK5E were synthesized using **3b** and the corresponding alcohols (HO-EG₃, FUDR-5′-TBS, and EdU-3′-TBS) (Fig. 4d). Please note that conditions for the TBS

deprotection were mild and compatible with the ketals, and no side products were observed in the last step in the synthesis of PTX-derived prodrugs. In addition, although ketal exchange reactions may occur and yield symmetric homodimers in synthesizing asymmetric ketal-based prodrugs, we did not note the formation of homodimers using our method. For PTX-7-K-EG₃ with a short PEG chain, PEGylation of PTX via an acid-sensitive acetone-based ketal, which would be traceless, might address the solubility problem of PTX without causing side effects associated with Cremophor EL—the excipient of a commercial PTX formulation (Taxol). For PK3F and PK5F, the drug FUDR was used to synthesize heterodimeric prodrugs to improve antitumor efficacy, and they might show different hydrolysis kinetics. For PK5E, 5-ethynyl-2′-deoxyuridine (EdU), a nucleoside analogue that is widely used for detecting live cells via a copper-catalyzed fluorogenic click reaction, can be used to investigate the intracellular hydrolysis of the heterodimers.

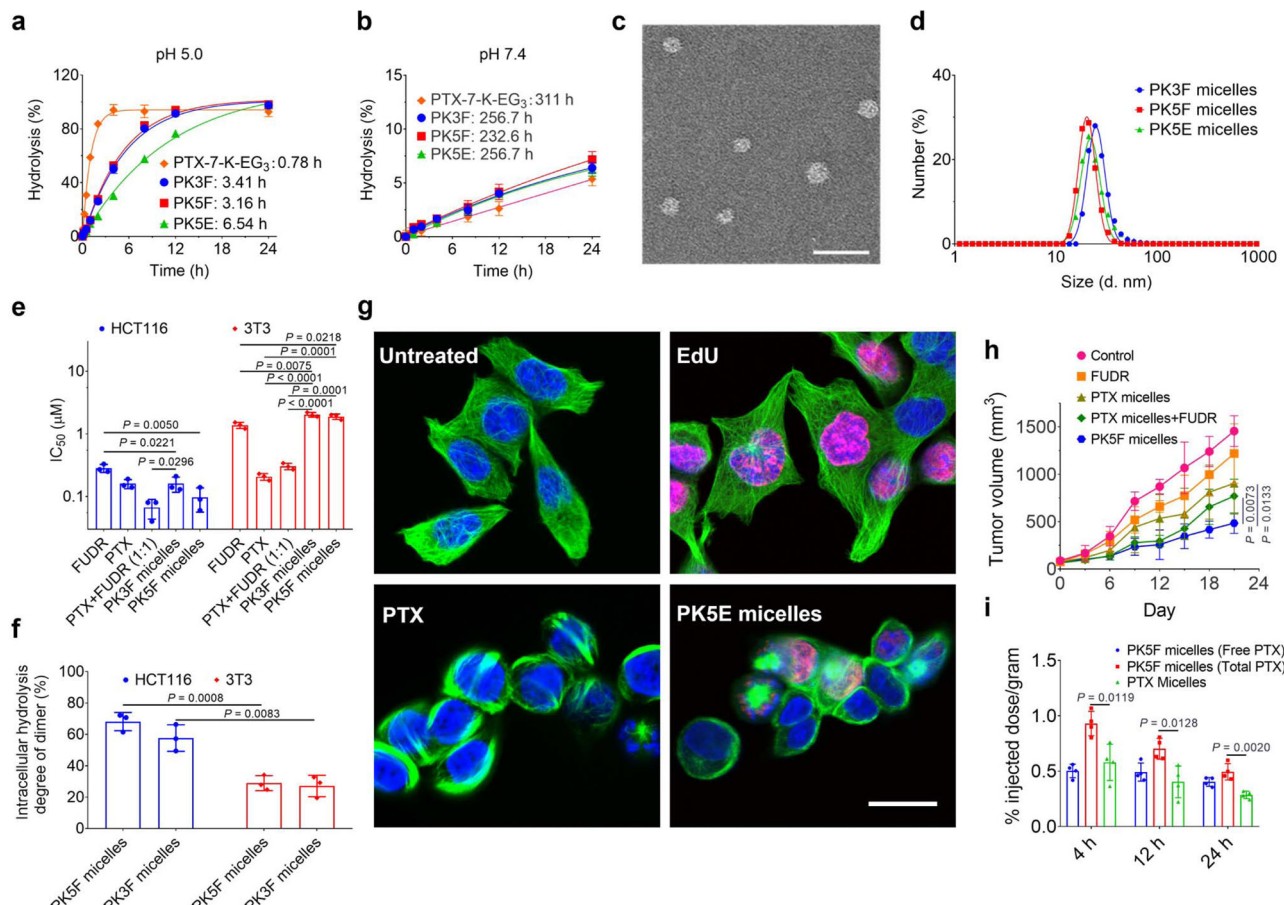

**Fig. 5 In vitro and in vivo characterizations of PTX prodrugs.** Hydrolysis of PTX prodrugs at (**a**) pH 5.0 and (**b**) pH 7.4. Data of (**a**) and (**b**) are means ± SD ($n = 4$ independent samples). **c** Representative transmission electron microscope image of PK5F-loaded micelles; scale bar = 50 nm. Experiments were performed three times independently, and one representative image is shown. **d** Sizes of heterodimer-loaded micelles, determined by dynamic light scattering. **e** $IC_{50}$ values of drugs against HCT116 cells and 3T3 cells. **f** The extent of intracellular hydrolysis (%) of heterodimers after incubation of heterodimer-loaded micelles with HCT116 cells and 3T3 cells at 20 μM for 6 h. Data of (**e**) and (**f**) are means ± SD ($n = 3$ independent experiments), and unpaired two-tailed Student's $t$-test was used for statistical analysis. **g** Confocal microscopy images of intracellular hydrolysis of PK5E. Untreated cells and cells incubated with EdU, PTX, and micelle-encapsulated PK5E at 5 μM for 6 h at 37 °C. Nuclei were stained with Hoechst 33342 (blue). $\alpha$-Tubulin was stained with $\alpha$-tubulin (11H10) Rabbit mAb (green). EdU was stained using Alexa Fluor 555 azide via click chemistry (pink). Scale bar = 20 μm. Experiments were performed three times independently, and representative images are shown. **h** Evaluation of in vivo efficacy of FUDR, PTX-loaded micelles, PTX-loaded micelles plus FUDR, and PK5F-loaded micelles against HCT116 xenograft tumors. Temporal dependence of tumor volume after intravenous injection of the four formulations on days 0, 3, 6, and 9 at the following doses: FUDR, 11.7 μmol/kg; PTX-loaded micelles, 11.7 μmol/kg; PTX-loaded micelles plus FUDR, 11.7 μmol/kg for each drug; PK5F-loaded micelles, 11.7 μmol/kg. Data are means ± SD ($n = 6$ biologically independent animals). **i** Accumulation (% injected dose/gram of tumor) of drugs at 4, 12, and 24 h after a single intravenous injection. Free PTX indicates the liberated PTX from PK5F. Data are means ± SD ($n = 4$ biologically independent animals). Unpaired two-tailed Student's $t$-test was used for statistical analysis in (**h**) and (**i**). Source data of (**a**), (**b**), (**e**), (**f**), (**h**) and (**i**) are provided as a Source Data file.

**In vitro and in vivo characterizations of PTX prodrugs.** Hydrolysis experiments indicated that hydrolysis of PTX prodrugs yielded native PTX (Supplementary Figs. 20–21), thus confirming that the acetone-based ketal in the prodrugs acted as a traceless linker. At pH 5.0 (lysosomal pH), the $t_{1/2}$ values (calculated from kinetic constant values; Supplementary Table 9) of PTX-7-K-EG$_3$, PK3F, PK5F, and PK5E were 0.78, 3.41, 3.16, and 6.54 h (Fig. 5a and Supplementary Fig. 22a), respectively; whereas, at pH 7.4 (physiological pH), the corresponding values exceeded 200 h (Fig. 5b and Supplementary Fig. 22b). Note that the $t_{1/2}$ of PK5F did not significantly differ from that of PK3F; although the fluorine atom is far from the ketal bond in the PKFs (i.e., PK3F and PK5F), the rate of hydrolysis of PK5E, which has an alkynyl group instead of a fluorine atom, was lower than that of PKFs. In addition, the $t_{1/2}$ values of these heterodimers were larger than the values of PTX-7-K-EG$_3$, probably because of the greater steric bulk of the promoieties in the former. The difference between the $t_{1/2}$ values of the same prodrug at the two pHs was not only related to the difference between the two pH values ($\Delta$pH), probably because ketal hydrolysis proceeds by general acid catalysis, the rate of which can be affected by multiple factors (e.g., the composition of the buffer)[57,61,62].

The use of PKFs for co-delivery and simultaneous release of PTX and FUDR—which have different tumor-inhibiting mechanisms and non-overlapping adverse effects—might result in improved anticancer efficacy than either agent alone[63]. We were unable to prepare excipient-free self-assembled prodrug nanoparticles, perhaps because paclitaxel is too hydrophobic. Thus, we encapsulated the PK5F, PK3F, and PK5E, which are hydrophobic and poorly water-soluble, in methoxy polyethylene glycol-poly(D,L-lactide) (mPEG-PDLLA, $M_n = 5000$–3000 g/mol) micelles (about 20 nm, drug loading: ~8.9 wt%, loading efficiency:

~74%) (Fig. 5c, d, Supplementary Table 10, and Supplementary Fig. 23), for following biological evaluations. The $IC_{50}$ values of micelle-encapsulated PK3F and PK5F in a human colon cancer cell line HCT116 were lower than that of PTX and FUDR, and were 159 and 90.4 nM, respectively (Fig. 5e and Supplementary Fig. 24). Even though the $t_{1/2}$ values for free PK5F and PK3F hydrolysis did not differ significantly, micelle-encapsulated PK5F exhibited higher toxicity than micelle-encapsulated PK3F, indicating that other processes, such as enzymatic hydrolysis, may have been involved, to different extents, in intracellular hydrolysis of the heterodimers. The degree of hydrolysis of micelle-encapsulated PK5F in HCT116 cells was 68% at 6 h, and the value for micelle-encapsulated PK3F was slightly lower (Fig. 5f and Supplementary Fig. 25). Intracellular hydrolysis of heterodimers was also examined using confocal microscopy using PK5E (Fig. 5g). Cells treated with micelle-encapsulated PK5E showed notable EdU staining and microtubule bundles, confirming that the extent of intracellular hydrolysis of micelle-encapsulated PK5E was high and that the released EdU was incorporated into the DNA of the proliferating cells. However, micelle-encapsulated PKFs displayed lower toxicities than PTX and FUDR in the healthy 3T3 cells (Fig. 5e), and the extent of hydrolysis of micelle-encapsulated PKFs was much lower in the 3T3 cells than in the HCT116 cells (Fig. 5f), perhaps because the two types of cells have different proliferation and metabolism modes. Our findings indicate that the heterodimers were activated more efficiently by tumor cells than by healthy cells.

In vivo anticancer efficacy study in an HCT116 xenograft model revealed that PK5F-loaded micelles, which had a lower $IC_{50}$ than PK3F-loaded micelles, exhibited better efficacy than PTX-loaded micelles plus FUDR (Fig. 5h and Supplementary Figs. 27–30), probably because PK5F-loaded micelles improved co-delivery of PTX and FUDR to the tumor. Although the carrier vehicle is the same for PTX and PK5F, PK5F showed higher accumulation in the tumor than did PTX at all time points (Fig. 5i). We speculate that different accumulation values of drugs may be ascribed to different compatibility between drug and mPEG-PDLLA, which influences complex interactions between biological components, drugs, and micelles[64]. The extent of PK5F hydrolysis in tumors was also determined, and the degree of hydrolysis had increased from 55% at 4 h to 82% at 24 h after intravenous injection, indicating that intratumoral hydrolysis of the heterodimer was efficient (Supplementary Fig. 26). We note that the hydrolysis rate of the prodrug in the tumor tissue was slower than the in vitro intracellular hydrolysis rate, possibly because the tumor's microenvironment is less acidic than endolysosomes[1]. Moreover, no significant loss of body weight was observed for any treatments (Supplementary Fig. 30). These results indicate that loading heterodimeric prodrugs into polymeric micelles can enable co-delivery of two drugs with different physicochemical properties and is a promising strategy for improved cancer therapy.

**Synthesis of ketal-linked biomaterials**. In addition to prodrugs, IPPEs could have tremendous utility in the development of novel biomaterials. For example, Fréchet et al. have modified dextran, a highly water-soluble natural biopolymer, as pH-sensitive and hydrophobic ketalated dextran using 2-MPE (**2a**, the simplest IPPE), and demonstrated their vast potential in biomaterials and drug delivery[11,12,65–68]. Nevertheless, the lack of diversity and availability of IPPEs limits the widespread applications of such ketalated dextran. To demonstrate the utility of IPPEs in creating functionalized ketalated dextran, IPPEs **3t** and **3r** were reacted with dextran ($M_w = 10,000$ g/mol) to afford acrylate- and alkyne-functionalized ketalated dextran derivatives (Fig. 6 and see details

in Supplementary Information), respectively. Ketalated dextran with mixed ketals of acyclic ketals and cyclic ketals was yielded because acyclic ketals were partially reacted with its neighboring hydroxyls to form more stable cyclic ketals through a ketal exchange reaction. However, the hydroxyl substitution degree, the ratio of acyclic ketals to cyclic ketals, and hydrolysis kinetics of ketalated dextran could be well-tuned by adjusting the reaction conditions (e.g., reaction time)[11]. The functionalized ketalated dextran derivatives would allow versatile modification on demand by Michael addition reaction or click reaction. In addition, the drug-derived IPPE **3k** was reacted with dextran to obtain a pH-sensitive dextran-drug conjugate.

In summary, we have developed an organocatalytic transiso-propenylation approach to synthesize IPPEs that are difficult to access so far, using Brønsted acid–base pair as the organocatalyst. To the best of our knowledge, we accomplished the first syntheses of drug-derived IPPEs. We used an IPPE derivative of PTX to demonstrate the late-stage derivatization of drug-derived IPPEs for implementing the synthesis of modular ketal-linked prodrugs, enabling the development of polymeric micelles loaded with a heterodimeric PTX/FUDR prodrug—which showed higher potent anticancer efficacy than PTX. The prodrug strategy reported herein opens up an avenue for the development of modular prodrugs from which active drugs can be released tracelessly upon exposure to acid. In addition, functionalized ketalated dextran derivatives were also readily obtained using the functional group-containing IPPEs. The straightforward synthesis and late-stage derivatization of IPPEs will facilitate the development and practical application of acid-sensitive prodrugs, drug delivery systems, and biomaterials.

## Methods

**Syntheses**. All details regarding the synthesis and characterization of IPPEs, relevant intermediates, prodrugs, and ketalated dextran derivatives are provided in the Supplementary Information.

**Hydrolysis of prodrugs determined using HPLC**. To investigate the hydrolysis kinetics, prodrugs were dissolved in ACN and diluted 19 times using buffers (30 mM, pH 5.0 acetate buffer, pH 7.4 phosphate buffer) to final concentration of 20 µM in 200 µL buffers of each sample. Hydrolysis experiments were performed in a constant temperature shaker (Zhicheng Inc. CHN) at 200 rpm and 37 ℃ ($n = 4$). At predetermined intervals, samples were added with 200 µL PB solution (200 mM, pH 8.0) and 400 µL ACN immediately and stored at −20 ℃ before measurement. The concentration of PTX was analyzed by HPLC. HPLC analyses were performed using Agilent 1260 instruments (Agilent, USA) with C18 column (Agilent, Poroshell 120 EC-C18, 4.6 mm × 100 mm, 2.7 µm). For PTX-7-K-EG$_3$, samples were injected at a flow rate of 1.0 mL/min, column temperature at 25 °C, and UV detection wavelength at 227 nm. An aqueous solution of ACN/H$_2$O (65/35) was used as the mobile phase. For dimeric prodrugs, samples were injected at a flow rate of 1.0 mL/ min, column temperature at 25 °C, and UV detection wavelength at 227 nm. HPLC eluent condition: 20% [v/v] ACN, 1.0 min 20% [v/v] ACN, 4.0 min 80% [v/v] ACN, 7.0 min 80% [v/v] ACN, 7.5 min 20% [v/v] ACN, 10 min 20% [v/v] ACN.

The hydrolysis (%) of prodrug was calculated by the following formula:

$$\text{Hydrolysis (\%)} = \frac{C_n}{C_{\text{total}}} \times 100\% \tag{1}$$

where $C_n$ represents the concentration of PTX in the $n$th sample and $C_{\text{total}}$ represents concentration of PTX after complete hydrolysis of prodrug at pH 3.0.

The data were fitted with "Nonlinear regression (curve fit)-One phase decay" model using GraphPad Prism 7 (GraphPad Software, CA), and the rate constant was determined by a "Linear regression" model of $\ln(100 - 100 \times C_n/C_{\text{total}})$ vs time. $t_{1/2} = \ln 2/k$.

**Preparation and characterization of drug-loaded micelles**. Nanoprecipitation technique was utilized to load drugs into mPEG-PDLLA micelles. Briefly, mPEG-PDLLA (10 mg) and drug (1 mg) was dissolved in 0.5 mL acetone. The solution was slowly dropped into 10 mL phosphate buffer (10 mM, pH 7.4) under stirring (500 rpm). The residual solvent was removed by evaporation at room temperature for 2 h. After that, drug-loaded micelles were filtrated with 0.45 µm membrane filter, lyophilized to dryness, and stored in the freezer before use.

The yield of drug-loaded micelles was weighed. The hydrodynamic diameter, polydispersity index, and zeta potential of drug-loaded micelles were analyzed by

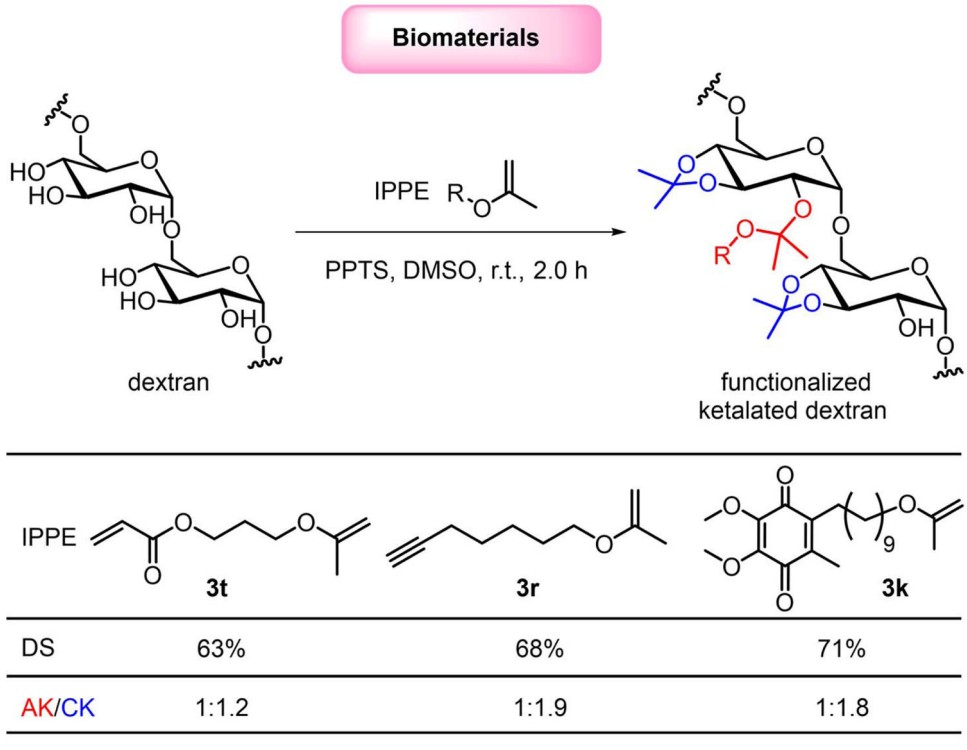

**Fig. 6 Ketal-linked biomaterials.** Synthesis of functionalized ketalated dextran, using addition reaction of IPPEs to hydroxyls of dextran under catalysis by pyridinium *p*-toluenesulfonate (PPTS). DS denotes the hydroxyl substitution degree; AK denotes the acyclic ketals; CK denotes the cyclic ketals.

DLS. Sizes of drug-loaded micelles were also observed by TEM (Talos F200C). The drug loading and loading efficiency of drug-loaded micelles were determined by HPLC. An aqueous solution of ACN/$H_2O$ (65/35) was used as the mobile phase. Drugs were detected by UV absorbance at $\lambda = 227$ nm and quantified using corresponding standard curves. The stability of drug-loaded micelles was studied using DLS and HPLC. The results showed that drug-loaded micelles were stable in the aqueous solution without prodrug hydrolysis and notable size changes in a week. In addition, the lyophilized drug-loaded micelles could be easily re-dispersed.

The drug loading (DL) and loading efficiency (LE) of drugs in micelles were calculated by the following formulas:

$$DL\,(wt\%) = \frac{\text{actual measured mass of drug}}{\text{mass of drug loaded micelles}} \times 100wt\% \quad (2)$$

$$LE\,(\%) = \frac{\text{actual measured loading of drug in drug loaded micelles}}{\text{theoretical (feed) loading of drug in drug loaded micelles}} \times 100\% \quad (3)$$

**Cell cytotoxicity assay.** Standard Cell Counting Kit-8 (CCK-8) assay was followed to evaluate the cytotoxicity of free drugs and micellar formulations or combinations (FUDR:PTX = 1:1, mol/mol) in HCT116 cells and 3T3 cells. Cells were seeded at a seeding density of 2500 cells/well in a 96 well plate, and cultured in 100 μL DMEM (or RPMI 1640 medium) containing 10% fetal bovine serum (FBS), 100 IU/mL penicillin, and 100 μg/mL streptomycin in 5% $CO_2$ incubator at 37 °C for 24 h ($n = 3$). PTX was dissolved in DMSO, whereas, FUDR and drug-loaded micelles were in PBS (6.7 mM, pH 7.4). The medium in each well was replaced by 100 μL fresh medium containing drug. For HCT116 cells, drugs were ranging from 1 nM to 5 μM; for 3T3 cells, drugs were ranging from 10 nM to 20 μM. The cells were further incubated at 37 °C for 72 h. The medium in each well was aspirated, and 100 μL fresh medium plus 10 μL CCK-8 solution was added, followed by incubation at 37 °C for another 1.5 h. Absorbance was measured by a SpectraMax i3x (Molecular Devices, CA) at 450 nm. The data were fitted with "Dose-Response – Inhibition" model using GraphPad Prism 7 (GraphPad Software, CA), and the $IC_{50}$ values were reported using equation of "log(inhibitor) vs. response—Variable slope (four parameters)". Experiments were performed in triplicate.

**Cellular uptake and intracellular hydrolysis studies.** HCT116 cells or 3T3 cells were seeded at a density of $2 \times 10^5$ cells in a six-well plate with DMEM medium containing 10% fetal bovine serum (FBS), 100 IU/mL penicillin, and 100 μg/mL streptomycin. Cells were grown in a 37 °C incubator with 5% $CO_2$ overnight to allow to attach ($n = 3$). Drugs with an equivalent concentration of 20 μM PTX were prepared as procedures above. The cell medium was replaced by 2 mL drug-containing medium. After 6 h incubation at 37 °C, the medium was aspirated and the cells were washed twice with cold PBS. Cells were de-attached from the six-well plate by trypsinization at room temperature. The cell pellets were collected by centrifugation at $150 \times g$ for 5 min. The cell pellets were resuspended by a mixture of ACN/$H_2O$ (1:1) and homogenized by sonication (VC 505, Vibracell Sonics, Newton, USA). Then, 500 μL of cell homogenates were deproteinized by the addition of three volumes of cold ACN which contains 25 μL docetaxel solution (4 μg/mL) as internal standard (IS). The mixture was centrifuged at $19,800 \times g$ for 10 min to remove precipitates. Three mL methyl tert-butyl ether was added to the supernatant to extract drugs, followed by vortex and centrifugation. The organic layer was aspirated and dried by an Eppendorf™ Concentrator Plus vacuum concentrator (Eppendorf, GER). The concentrations of PTX and dimers were analyzed by HPLC as mentioned above.

To investigate intracellular hydrolysis and distribution of PK5E, HCT116 cells were seeded onto 13-mm coverslips ($5 \times 10^4$ cells per coverslip) in 24-well plate with DMEM medium containing 10% fetal bovine serum (FBS), 100 IU/mL penicillin and 100 μg/mL streptomycin. Cells were grown in a 37 °C incubator with 5% $CO_2$ overnight to allow to attach. The cell medium was replaced by 500 μL drug-containing medium (5 μM). After 6 h incubation at 37 °C, the medium was removed, and the cells were washed twice with PBS and fixed with 4% paraformaldehyde solution. Following standard protocols, cells were first stained using BeyoClick™ EdU-555 Kit (Beyotime Biotechnology, CHN) and then stained with α-Tubulin (11H10) Rabbit mAb (Alexa Fluor® 488 Conjugate) (Cell Signaling Technology, USA). The nucleus was stained with Hoechst 33342 solution (Beyotime Biotechnology, CHN). Slides were sealed using Vectashield mounting medium (Vector Laboratories Burlingame, CA, USA) and observed using CLSM (Leica Microsystems, Wetzlar, GER).

**In vivo antitumor evaluation.** All experiments were carried out following the guidelines of the Beijing Laboratory Animal Center, and approved by the Ethical Commission at Nankai University. Female BALB/c nude mice (18–20 g) were purchased from Beijing Vital River Laboratory Animal Technology Co., Ltd. and maintained in a 12 h light-dark cycle at 25 °C, 40% relative humidity with free access to food and water. Five million HCT116 cells in 100 μL DMEM medium were subcutaneously injected into the right flank of mice. After the tumor volume reached ~100 mm³, mice were randomly divided into five groups ($n = 6$). HCT116 tumor-bearing mice were intravenously injected with PBS, FUDR, PTX-loaded micelles, PTX-loaded micelles plus FUDR and PK5F-loaded micelles at a dose of 11.7 μmol/kg at day 0, 3, 6, and 9. The weight and tumor volume of each mice were measured every three days. The volume of the tumor was calculated using the formula: $V = (\text{length} \times \text{width}^2)/2$. Three weeks post injection, mice were sacrificed. The blood was drawn for blood count test and serum biochemical analysis. Major organs and tumors were collected, followed by fixed in 4% paraformaldehyde solution and embedded in paraffin. Organ tissues were sectioned for H&E staining, and tumor tissues were sectioned for H&E staining, Ki67 immunohistochemical staining, and TUNEL assay.

HCT116 tumor-bearing nude mice ($n = 4$) were received single injection of PTX-loaded micelles and PK5F-loaded micelles at the dose of 11.7 μmol/kg respectively. At predetermined time intervals (4, 12, and 24 h), mice were sacrificed and their tumors were dissected to measure accumulation of PTX and PK5F and hydrolysis degree of PK5F by HPLC.

**Statistic analysis**. Data were reported as means with standard derivations. Statistical analysis was conducted by Student's $t$-test using GraphPad Prism version 7 for Windows (GraphPad Software, CA). Differences were considered statistically significant if $p < 0.05$.

**Reporting summary**. Further information on research design is available in the Nature Research Reporting Summary linked to this article.

## Data availability

The source data underlying Fig. 5a–b, e–f, h–i, Supplementary Figs. 22, 24–26 and 29–30 are provided as a Source Data file. All data supporting the findings of this study are available within the article and the Supplementary Information, and are also available from the corresponding author upon reasonable request. Source data are provided with this paper.

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

## Acknowledgements

This study was supported by the National Natural Science Foundation of China (51773098), China Postdoctoral Science Foundation (2021M690793), Natural Science Foundation of Tianjin City of China (18JCYBJC28300), NIH R01 GM131728, and Fundamental Research Funds for Central Universities (China). The authors would like to thank Professors Qi-Lin Zhou, Shou-Fei Zhu, and Mengchun Ye at Nankai University for their helpful discussions. This work is dedicated to the 100th anniversary of Chemistry at Nankai University.

## Author contributions

N.Y., Y.X., D.S.K., and S.G. conceived and designed the study. N.Y., Y.X., T.L., H. Zhong., Z.X., H. Zou., J.M., and Z.C. performed the research. N.Y., Y.X., T.L., T.J., X.-J. L., L.S., D.S.K., and S.G. analyzed the data and wrote the manuscript. All authors contributed to the manuscript.

## Competing interests

The authors declare no competing interests.
