## [Peer Review File · Nature Communications]

REVIEWER COMMENTS

Reviewer #1 (Remarks to the Author):

The work of Guo and coworkers describes the development of a method for the transisopropenylation of a range of alcohols in an organocatalytic process with mild reaction conditions. The authors were able to show the versatility of this process for synthesising a range of prodrugs featuring acid-labile ketals, and finally the utility of this method through in vitro and in vivo analysis of coupled drugs. Overall this is a really nice study and I think will be relevant for many researchers working with acid-responsive polymers and drug delivery systems, and I recommend acceptance after some minor revisions have been addressed.

The section on synthetic scope was a little brief for me, and I was confused when trying to match this up with the supporting information, as Section 4 is extremely long and Section 7 quite a bit after. However, when reading the text, it jumps straight from section 4 to synthesising the TBS protected PTX derivatives. I think this section would be clearer if the process was more explicitly stated, ie. several drugs could not be selectively converted to the IPPE product, instead formed a mixture of MOP and IPPE, and to address this, TBS protection of the alcohol was first done followed by the selective IPPE. I also would like to see an explicit description that the conditions for the TBS deprotection are compatible with the presence of the ketal with no side products – I assume this is true based on reactions in the supporting information, but it would help readers to be able to see this clearly in the main text.

Figure 2 should be expanded to also provide reaction schemes, even just a representative one for HAS-K-DEX, in the main text the reaction is described but it's hard to follow without having to flick back and forth through the paper and the SI. I found the majority of the text describing Figure 2 to be a bit hard to follow and think the figure could definitely help with this with some revisions.

The cell biology is nicely done, and shows proof-of-concept that the drugs maintain activity, as well once encapsulated inside micelles. I would like to see statistics of the data in Figure 3e, the authors mentioned the particles have lower toxicity than the drugs in 3T3 cells, this isn't surprising but the stats on that graph would be good. I do have a major question about the in vivo work though, and that is why the PTX micelles have lower accumulation in the tumour than the PK5F micelles? If the only difference is the encapsulated drug, then I could understand why efficacy was different, but not why uptake in the tumour is different, considering the carrier vehicle is the same for both. The authors should comment on this, because if the PTX-micelles don't go into the tumour as effectively as the PK5F ones do, then the difference in efficacy is meaningless.

Reviewer #2 (Remarks to the Author):

Isopropenyl ethers are critical intermediates for the synthesis of medically valuable ketals but are challenging to prepare. The authors describe a straightforward and unprecedented organocatalytic transisopropenylation approach to synthesize isopropenyl ethers of alcohols. In a single procedure, the reactions proceed by the in situ formation of intermediate methoxy ketal, which then undergoes the elimination of methanol to produce isopropenyl ether. A cheap and readily available organocatalyst is used to catalyze transisopropenylation under a mild reaction condition efficiently. The authors have demonstrated the broad scope of the reaction thoroughly and have conducted some control experiments to support the proposed plausible reaction mechanism. Notably, the method is applicable to synthesize isopropenyl ethers from complex substrates (e.g., paclitaxel and floxuridine) that could not be obtained using previous conditions. Overall, the technique is demanded and practical. The authors have also comprehensively demonstrated the derivatization of isopropenyl ethers to synthesize acid-sensitive modular prodrugs and biomaterials and elucidated the possibility of

anticancer prodrugs in improving anticancer efficacy. Interestingly, they applied their chemistry to conjugate two drug molecules (paclitaxel and floxuridine) with distinct anticancer mechanisms to enable the co-delivery and co-release of the two drugs using micelles for improved therapy. Considering the high quality of the results during the chemistry, characterization, hydrolysis kinetics, cell experiments, and in vivo studies, this manuscript can bring new perspectives in the field of medicine and biomaterials. The work is well done, and the manuscript is well written. On this basis, the reviewer highly recommends its publication after the minor revision noted below.

Minor comments:

1. As a ketal exchange reaction might occur, are symmetric drug dimers found in the preparation of modular ketal-linked prodrugs?
2. Floxuridine, a hydrophilic drug, has been conjugated with some hydrophobic drugs, such as camptothecin and bendamustine, for combination therapy. It has been reported that some amphiphilic twin drugs can self-assemble into stable and carrier-free nanoparticles. How about heterodimers of paclitaxel and floxuridine? Can they self-assemble into stable nanoparticles? This needs to be discussed in the manuscript.
3. The drug loading of PK5F in mPEG-PDLLA micelles was ~8.9%. Have the authors attempted to increase the drug loading content?
4. PK5F seems to be a promising anticancer entity. However, the cost for the synthesis of PK5F using PTX-IPPE as an intermediate would be higher than using FUDR-IPPE. Is it possible to synthesize PK5F using FUDR-IPPE?
5. Have the authors investigated the stability and storage condition of the PK5F-loaded micelle? Please describe this in the experiment section.
6. Why are cyclic ketals formed in the synthesis of dextran derivatives? Please explain the mechanism of the formation of cyclic ketals during the synthesis of dextran derivatives.
7. The hydrolysis rate of PK5F-loaded in micelle in cell culture experiment (Figure 3f) seems to be faster than that in tumor (Supplementary Figure 26). Please discuss it in the manuscript.
8. A critical application of isopropenyl ethers is for the synthesis of acid-sensitive biomaterials. Thus, the description of modular ketal biomaterials should be included in the conclusion.
9. The molecular weight of mPEG-PDLLA should be clearly described.
10. The scale bars are missed in Figure 3g and Supplementary Figures 27 and 28.
11. The recently published review or research articles should be discussed in the revision, for example, Nano Today 2019, 29, 100800.

Typo errors:

1. In the main text, "to differing extents" should be written as "to different extents".
2. Reference 2 lacks the number of volume and pages.

Responses to reviewers' comments

Reviewer #1 (Remarks to the Author):

The work of Guo and coworkers describes the development of a method for the transisopropenylation of a range of alcohols in an organocatalytic process with mild reaction conditions. The authors were able to show the versatility of this process for synthesising a range of prodrugs featuring acid-labile ketals, and finally the utility of this method through in vitro and in vivo analysis of coupled drugs. Overall this is a really nice study and I think will be relevant for many researchers working with acid-responsive polymers and drug delivery systems, and I recommend acceptance after some minor revisions have been addressed.

Response: We thank the reviewer very much for the positive assessments and appreciation. We would also like to thank the reviewer for the comments below, which helped us to improve the quality of our work.

The section on synthetic scope was a little brief for me, and I was confused when trying to match this up with the supporting information, as Section 4 is extremely long and Section 7 quite a bit after. However, when reading the text, it jumps straight from section 4 to synthesising the TBS protected PTX derivatives. I think this section would be clearer if the process was more explicitly stated, ie. several drugs could not be selectively converted to the IPPE product, instead formed a mixture of MOP and IPPE, and to address this, TBS protection of the alcohol was first done followed by the selective IPPE.

Response: We are sorry for the confusion. To make the section clearer, two major changes were done in the revised section on synthetic scope and listed below.

"As PTX could not be selectively converted to the diIPPE product, instead a mixture of MOP and IPPE, 2'-hydroxyl of PTX was first protected by TBS (tert-butyldimethylsilyl), and then 7-hydroxyl of PTX was selectively transisopropenylated. As a result, we successfully synthesized PTX-2'-TBS-7-IPPE (**3b**) from PTX-2'-TBS with a yield of 85%."

"Similar to PTX, FUL could not be selectively converted to the diIPPE product. Therefore, TBS protection of FUL at 3-position was first done, and then FUL-3-TBS-17-IPPE (**3c**) was synthesized from FUL-3-TBS."

I also would like to see an explicit description that the conditions for the TBS deprotection are compatible with the presence of the ketal with no side products – I assume this is true based on reactions in the supporting information, but it would help readers to be able to see this clearly in the main text.

Response: The reviewer's assumption is correct. The conditions for the TBS deprotection in our study were mild and compatible with the ketals.

An explicit description was added in the revised manuscript: "To demonstrate the utility of drug-derived IPPEs for the modular preparation of ketal-linked prodrugs, we further synthesized four PTX prodrugs, designated PK5F, PTX-7-K-EG₃, PK3F, and PK5E (**Figure 2c, d**; see Supplementary Section 9). As shown in **Figure 2c**, PK5F was synthesized using PPTS-catalyzed reaction of PTX-2'-TBS-7-IPPE **3b** with FUDR-3'-TBS and subsequent removal of the TBS group with tetra-n-butylammonium fluoride (TBAF). Similarly, PTX-7-K-EG₃, PK3F, and PK5E were synthesized using **3b** and the corresponding alcohols (HO-EG₃, FUDR-5'-TBS, and EdU-3'-TBS) (**Figure 2d**).

Please note that conditions for the TBS deprotection were mild and compatible with the ketals, and no side products were observed in the last step in the synthesis of PTX-derived prodrugs. In addition, although ketal exchange reactions may occur and yield symmetric homodimers in synthesizing asymmetric ketal-based prodrugs, we did not note the formation of homodimers using our method."

Figure 2 should be expanded to also provide reaction schemes, even just a representative one for HAS-K-DEX, in the main text the reaction is described but it's hard to follow without having to flick back and forth through the paper and the SI. I found the majority of the text describing Figure 2 to be a bit hard to follow and think the figure could definitely help with this with some revisions.

Response: We thank the reviewer for the suggestions and have redrawn **Figure 2**. In the revised **Figure 2**, we showed two representative reaction schemes.

Revised **Figure 2. Ketal-linked prodrugs.** (a) synthesis of PEG-K-DEX; (b) structures of LA-K-CAPME, LA-K-BUF, and HSA-K-DEX; (c) synthesis of PK5F; (d) structures of PTX-7-K-EG₃,

PK3F, and PK5E.

Accordingly, we also made some changes in the revised text to make the section easier to follow. For example:

"Given that IPPEs could readily react with hydroxyl-group-containing drugs to afford acyclic-ketal-linked prodrugs and having established the organocatalytic transisopropenylation method for synthesis of IPPEs, we further explored the utility of this method for accessing divergent asymmetric ketal-linked prodrugs (**Figure 2** and see Supplementary Section 9). As proof-of-principle, hydroxyl-group-containing dexamethasone (DEX), a hydrophobic drug, was reacted with a PEG-derived IPPE **3m** ($M_n = 2,000$ g/mol) under catalysis by dichloroacetic acid (DCA) to obtain a water-soluble *O,O*-ketal-linked PEGylated prodrug, *i.e.*, PEG-K-DEX (**Figure 2a**). In addition to *O,O*-ketal-linked prodrug, an *O,S*-ketal-linked prodrug and an *O,ON*-ketal-linked prodrug, which both are rarely reported and may have different stimuli-responsiveness than *O,O*-ketals,⁵⁸⁻⁶⁰ were also enabled by using a lauryl alcohol-derived IPPE under similar reaction conditions. Specifically, an *O,S*-ketal prodrug of thiol-group-containing captopril methyl ester (CAPME) and an *O,ON*-ketal prodrug of hydroxylamine-group-containing bufexamac (BUF) (**Figure 2b**) were synthesized. Moreover, IPPE **3w** was similarly reacted with hydroxyl-group-containing DEX to obtain an intermediate, which was subsequently conjugated to human albumin serum (HSA) through maleimide-thiol reaction to yield a water-soluble *O,O*-ketal-linked protein prodrug conjugate, *i.e.*, HSA-K-DEX (**Figure 2b**)."

"To demonstrate the utility of drug-derived IPPEs for the modular preparation of ketal-linked prodrugs, we further synthesized four PTX prodrugs, designated PK5F, PTX-7-K-EG₃, PK3F, and PK5E (**Figure 2c, d**; see Supplementary Section 9). As shown in **Figure 2c**, PK5F was synthesized using PPTS-catalyzed reaction of PTX-2'-TBS-7-IPPE **3b** with FUDR-3'-TBS and subsequent removal of the TBS group with tetra-*n*-butylammonium fluoride (TBAF). Similarly, PTX-7-K-EG₃, PK3F, and PK5E were synthesized using **3b** and the corresponding alcohols (HO-EG₃, FUDR-5'-TBS, and EdU-3'-TBS) (**Figure 2d**). Please note that conditions for the TBS deprotection were mild and compatible with the ketals, and no side products were observed in the last step in the synthesis of PTX-derived prodrugs. In addition, although ketal exchange reactions may occur and yield symmetric homodimers in synthesizing asymmetric ketal-based prodrugs, we did not note the formation of homodimers using our method."

The cell biology is nicely done, and shows proof-of-concept that the drugs maintain activity, as well once encapsulated inside micelles. I would like to see statistics of the data in Figure 3e, the authors mentioned the particles have lower toxicity than the drugs in 3T3 cells, this isn't surprising but the stats on that graph would be good.

Response: The statistics of the data have been added in the revised **Figure 3e**.

I do have a major question about the in vivo work though, and that is why the PTX micelles have lower accumulation in the tumour than the PK5F micelles? If the only difference is the encapsulated drug, then I could understand why efficacy was different, but not why uptake in the tumour is different, considering the carrier vehicle is the same for both. The authors should comment on this, because if the PTX-micelles don't go into the tumour as effectively as the PK5F ones do, then the difference in efficacy is meaningless.

Response: We thank the reviewer for this outstanding question. The micelles' delivery could be affected by complex interactions between biological components, drugs, and micelles, so the delivery efficiency

for different drugs could be different and largely dependent on the drug-polymer compatibility, which can be theoretically qualified by the Flory–Huggins interaction parameters (please see: Letchford K, Liggins R, Burt H. Solubilization of hydrophobic drugs by methoxy poly (ethylene glycol)-block-polycaprolactone diblock copolymer micelles: Theoretical and experimental data and correlations. *Journal of Pharmaceutical Sciences*, 2008, 97(3): 1179-1190.).

We commented in the revised manuscript: "Although the carrier vehicle is the same for PTX and PK5F, PK5F showed higher accumulation in the tumor than did PTX at all time points (**Figure 3i**). We speculate that different accumulation values of drugs may be ascribed to different compatibility between drug and mPEG-PDLLA, which influences complex interactions between biological components, drugs, and micelles.⁶⁴".

Reviewer #2 (Remarks to the Author):

Isopropenyl ethers are critical intermediates for the synthesis of medically valuable ketals but are challenging to prepare. The authors describe a straightforward and unprecedented organocatalytic transisopropenylation approach to synthesize isopropenyl ethers of alcohols. In a single procedure, the reactions proceed by the in situ formation of intermediate methoxy ketal, which then undergoes the elimination of methanol to produce isopropenyl ether. A cheap and readily available organocatalyst is used to catalyze transisopropenylation under a mild reaction condition efficiently. The authors have demonstrated the broad scope of the reaction thoroughly and have conducted some control experiments to support the proposed plausible reaction mechanism. Notably, the method is applicable to synthesize isopropenyl ethers from complex substrates (e.g., paclitaxel and floxuridine) that could not be obtained using previous conditions. Overall, the technique is demanded and practical.

The authors have also comprehensively demonstrated the derivatization of isopropenyl ethers to synthesize acid-sensitive modular prodrugs and biomaterials and elucidated the possibility of anticancer prodrugs in improving anticancer efficacy. Interestingly, they applied their chemistry to conjugate two drug molecules (paclitaxel and floxuridine) with distinct anticancer mechanisms to enable the co-delivery and co-release of the two drugs using micelles for improved therapy. Considering the high quality of the results during the chemistry, characterization, hydrolysis kinetics, cell experiments, and in vivo studies, this manuscript can bring new perspectives in the field of medicine and biomaterials. The work is well done, and the manuscript is well written. On this basis, the reviewer highly recommends its publication after the minor revision noted below.

Response: We thank the reviewer very much for these positive comments on our work and the comments below, all of which helped us to improve the quality of our work.

Minor comments:

1. As a ketal exchange reaction might occur, are symmetric drug dimers found in the preparation of modular ketal-linked prodrugs?

Response: We thank the reviewer for this excellent question. We were aware of this possibility and carefully monitored the reactions using TLC and ¹H NMR. We did not see symmetric drug dimers in the isolated compounds.

We have added in the revised manuscript: "In addition, although ketal exchange reactions may occur and yield symmetric homodimers in synthesizing asymmetric ketal-based prodrugs, we did not note the

formation of homodimers using our method."

2. Floxuridine, a hydrophilic drug, has been conjugated with some hydrophobic drugs, such as camptothecin and bendamustine, for combination therapy. It has been reported that some amphiphilic twin drugs can self-assemble into stable and carrier-free nanoparticles. How about heterodimers of paclitaxel and floxuridine? Can they self-assemble into stable nanoparticles? This needs to be discussed in the manuscript.

Response: Preparation of excipient-free prodrug nanoparticles was explored. However, we failed to obtain stable self-assembled prodrug nanoparticles, perhaps because paclitaxel is too hydrophobic.

We have added in the revised manuscript: "We were unable to prepare excipient-free self-assembled prodrug nanoparticles, perhaps because paclitaxel is too hydrophobic."

3. The drug loading of PK5F in mPEG-PDLLA micelles was ~8.9%. Have the authors attempted to increase the drug loading content?

Response: We prepared micelles with a drug loading of 20% and 28.6% and found they were heterogeneous and easily aggregated. Thus, we used micelles with a drug loading of 8.9% in our study.

4. PK5F seems to be a promising anticancer entity. However, the cost for the synthesis of PK5F using PTX-IPPE as an intermediate would be higher than using FUDR-IPPE. Is it possible to synthesize PK5F using FUDR-IPPE?

Response: We thank the reviewer for the suggestion and had confirmed that PK5F could be synthesized using FUDR-IPPE. As we wanted to show the utility of PTX-IPPE for modular preparation of ketal-based prodrugs, thus the synthetic method of PK5F using FUDR-IPPE was not reported here.

5. Have the authors investigated the stability and storage condition of the PK5F-loaded micelle? Please describe this in the experiment section.

Response: We did study the stability of PK5F-loaded micelles using DLS and HPLC. The results showed that PK5F-loaded micelles with a drug loading of 8.9% were stable in the aqueous solution without prodrug hydrolysis and notable size changes in a week. In addition, PK5F-loaded micelles could be lyophilized and easily re-dispersed. Therefore, PK5F-loaded micelles were stored as a powder in the freezer before use. This has been described in the revised Methods section.

6. Why are cyclic ketals formed in the synthesis of dextran derivatives? Please explain the mechanism of the formation of cyclic ketals during the synthesis of dextran derivatives.

Response: The formation of cyclic ketals is a common phenomenon in the synthesis of acetalated dextran and has been well-documented in previous works (please see: Broaders K E, Cohen J A, Beaudette T T, et al. Acetalated dextran is a chemically and biologically tunable material for particulate immunotherapy. Proceedings of the National Academy of Sciences, 2009, 106(14): 5497-5502.). At the beginning of the reaction, acyclic ketals were preferably formed by isopropenyl ethers and hydroxyls. As the reaction proceeded, the more stable cyclic ketal was favorably formed through a ketal exchange reaction between acyclic ketal and its neighboring hydroxyl. This has been discussed in the revised manuscript.

The discussion on the mechanism was added in the revised manuscript: "Ketalated dextran with

mixed ketals of acyclic ketals and cyclic ketals was yielded because acyclic ketals were partially reacted with its neighboring hydroxyls to form more stable cyclic ketals through a ketal exchange reaction."

7. The hydrolysis rate of PK5F-loaded in micelle in cell culture experiment (Figure 3f) seems to be faster than that in tumor (Supplementary Figure 26). Please discuss it in the manuscript.

Response: *In vitro* hydrolysis study, we incubated tumor cells with PK5F-loaded micelles and removed the uninternalized PK5F-loaded micelles from the culture medium before determining the intracellular hydrolysis degree of the prodrug. In other words, the *in vitro* hydrolysis of prodrug mainly occurred inside tumor cells, likely in acidic endolysosomes. In contrast, hydrolysis of the prodrug in the tumor tissue occurred intracellularly and extracellularly. Although the tumor's microenvironment is slightly acidic, its acidity is weaker than endolysosomes. In addition, there could be other unknown factors (such as enzymes) affecting the hydrolysis and metabolism of the prodrug.

We have added in the revised manuscript: "We note that the hydrolysis rate of the prodrug in the tumor tissue was slower than the *in vitro* intracellular hydrolysis rate, possibly because the tumor's microenvironment is less acidic than endolysosomes.¹".

8. A critical application of isopropenyl ethers is for the synthesis of acid-sensitive biomaterials. Thus, the description of modular ketal biomaterials should be included in the conclusion.

Response: We have added in the revised conclusion: "In addition, functionalized ketalated dextran derivatives were also readily obtained using the functional group-containing IPPEs."

9. The molecular weight of mPEG-PDLLA should be clearly described.

Response: The molecular weight of mPEG-PDLLA was described in the original supporting information, and we have shown it in the revised manuscript.

10. The scale bars are missed in Figure 3g and Supplementary Figures 27 and 28.

Response: In revised **Figure 3g** and Supplementary **Figures 27** and **28**, the scale bars have been added.

11. The recently published review or research articles should be discussed in the revision, for example, Nano Today 2019, 29, 100800.

Response: We have referred relevant works in the discussion of the revised manuscript.

Typo errors:

1. In the main text, "to differing extents" should be written as "to different extents".

Response: This error has been corrected in the revised manuscript.

2. Reference 2 lacks the number of volume and pages.

Response: The number of volume and pages of Reference 2 have been updated in the revised manuscript.

REVIEWERS' COMMENTS

Reviewer #1 (Remarks to the Author):

I am satisfied that the reviewers have address all of my raised points. I remain unconvinced about the reasoning for the different in vivo activities of the micelles, however I do accept that this would be a complicated process to clarify and is not within the scope of this work.

Reviewer #2 (Remarks to the Author):

The revision is ready for publication.

Responses to reviewers' comments

Reviewer #1 (Remarks to the Author):

I am satisfied that the reviewers have address all of my raised points. I remain unconvinced about the reasoning for the different in vivo activities of the micelles, however I do accept that this would be a complicated process to clarify and is not within the scope of this work.

Response: We thank the reviewer for the positive comments.

Reviewer #2 (Remarks to the Author):

The revision is ready for publication.

Response: We thank the reviewer for the positive comments.